# Biomass Derived N-Doped Porous Carbon Made from Reed Straw for an Enhanced Supercapacitor

**DOI:** 10.3390/molecules28124633

**Published:** 2023-06-08

**Authors:** Yuyi Liao, Zhongtao Shang, Guangrui Ju, Dingke Wang, Qiao Yang, Yuan Wang, Shaojun Yuan

**Affiliations:** Low-Carbon Technology & Chemical Reaction Engineering Lab, College of Chemical Engineering, Sichuan University, Chengdu 610065, China; yimilyymail@163.com (Y.L.); taozi11281125@163.com (Z.S.); jugr0013@163.com (G.J.); dingkew0506@163.com (D.W.); yangqiaomail@163.com (Q.Y.)

**Keywords:** biomass, supercapacitors, nitrogen doping, melamine, porous carbon

## Abstract

Developing advanced carbon materials by utilizing biomass waste has attracted much attention. However, porous carbon electrodes based on the electronic-double-layer-capacitor (EDLC) charge storage mechanism generally presents unsatisfactory capacitance and energy density. Herein, an N-doped carbon material (RSM-0.33-550) was prepared by directly pyrolyzing reed straw and melamine. The micro- and meso-porous structure and the rich active nitrogen functional group offered more ion transfer and faradaic capacitance. X-ray diffraction (XRD), Raman, scanning electron microscopy (SEM), X-ray photoelectron spectroscopy (XPS), Brunauer–Emmett–Teller (BET) measurements were used to characterize the biomass-derived carbon materials. The prepared RSM-0.33-550 possessed an N content of 6.02% and a specific surface area of 547.1 m^2^ g^−1^. Compared with the RSM-0-550 without melamine addition, the RSM-0.33-550 possessed a higher content of active nitrogen (pyridinic-N) in the carbon network, thus presenting an increased number of active sites for charge storage. As the anode for supercapacitors (SCs) in 6 M KOH, RSM-0.33-550 exhibited a capacitance of 202.8 F g^−1^ at a current density of 1 A g^−1^. At a higher current density of 20 A g^−1^, it still retained a capacitance of 158 F g^−1^. Notably, it delivered excellent stability with capacity retention of 96.3% at 20 A g^−1^ after 5000 cycles. This work not only offers a new electrode material for SCs, but also gives a new insight into rationally utilizing biomass waste for energy storage.

## 1. Introduction

Currently, the world is facing a series of problems caused by fossil fuel combustion [1]. There is an urgent need to explore environmentally friendly and sustainable energy resources (wind, solar, and tidal energy, etc.) for energy requirements [2,3]. However, these sustainable energies are difficult to use directly on a large-scale due to their regional, unstable, and intermittent features [3]. The development of energy storage devices (EESs) such as high-power electrochemical capacitors and high-energy ion batteries utilizing clean and sustainable energy has attracted much attention. For batteries, the charge storage mechanism involves ion transport and redox reactions in bulk phase, thus providing a high energy density [4]. As for electrochemical capacitors, the charge storage mechanism involves the ion adsorption/desorption and insertion/deinsertion at the surface or near surface of the electrode, which can give a high power response [5]. Electric double-layer capacitors (EDLCs) exhibit high specific power (up to 10 KW kg^−1^), fast charge/discharge capability, and long cycle lives (e.g., millions of cycles) and have attracted much attention [6,7]. However, EDLCs contain porous carbon electrodes that use the charge storage mechanism of ion adsorption at the electrolyte and electrode interface, and no electrochemical reactions are involved [8,9]. This is responsible for the poor energy content of EDLCs [10]. Therefore, it is necessary to improve the porous carbon material to enhance the faradaic reaction for promoting the specific capacitance.

As a result of their environmental friendliness, abundance, and low cost in nature, renewable resources have attracted much attention for the preparation of electrode materials along with sustainable development [11]. Significant studies have been conducted on the subject of using lignin, bacterial cellulose, and cellulose to fabricate biomass-based electrode materials [12]. Biomass carbon material is a carbon-rich solid produced as a by-product from the thermochemical pyrolysis of biomass in an oxygen-limiting environment, which can serve as the electrode for supercapacitors (SCs) [13,14,15]. The chemical and physical properties of biomass carbon vary significantly with the feedstock and fabrication conditions [16,17]. Their carbonized network can provide more surface area and provide an ion diffusion pathway for charge storage. Momodu et al. synthesized activated carbon derived from bark through a simple activation and carbonization process, which showed a capacitance of 191 F g^−1^ at 1 A g^−1^ [18]. Kesavan et al. fabricated nitrogen-doped carbon nanosheets using peanut shell as a precursor. At a current density of 1 A g^−1^, a capacitance of 195 F g^−1^ could be achieved [19]. An Alpinia officinalis leaf derived porous carbon was reported by Taer et al., and it delivered a capacitance of 161 F g^−1^ at 1 A g^−1^ [20]. Kang et al. prepared hierarchical porous carbon materials using peanut bran as raw material through hydrothermal carbonization and a KOH activation reaction, and a capacitance of 188 A g^−1^ was reached at 0.04 A g^−1^ [21]. In comparison with other biomass waste such as peel, shell, bacteria, etc., reed straw is rich in cellulose, lignin, and hemicellulose [22]. In addition, about 900 million tons of corn, wheat, rice, and sorghum straw are produced every year in China, which can provide sustainable and abundant raw materials for preparing active carbon. The reed straw derived porous carbon is considered as a promising candidate for charge storage [23]. Such a hollow structure can provide more solid–liquid interface for EDLC. Dai et al. fabricated a hollow activated carbon with through-pore structure derived from reed straw. After the activation process, the prepared carbon electrode possessed a high specific surface area of 2387 m^2^ g^−1^ and a capacitance of 290 F g^−1^ [22]. Xie et al. prepared a porous carbon composite by combining raw reed straw with graphene nanosheets. The prepared composite electrode had a capacitance of 297 F g^−1^ at 0.1 A g^−1^ and a good cycling stability with a retention rate of 90% after 6000 cycles [23].

Until now, various natural biomass materials such as bacterial cellulose [24], orange peels [25], rice stems [26], bamboo [27], and carb shell [28] have been reported as the precursors to prepared carbon anodes for SCs. These biomass carbon materials based on EDLC charge storage mechanisms can be improved by increasing their surface area. However, the increased surface area contributing to EDLC is limited, and the capacitance and energy density of carbon materials usually remain poor [11]. Recently, the introduction of electrochemical-active functionalities (e.g., O, N, P, and S containing groups) on the carbon skeleton turned out to be a feasible approach to significantly boost the capacitance of carbon-based SCs [29,30,31,32]. These heteroatoms can expand the carbon interlayer distance and allow more transfer of anions and cations. Especially, nitrogen doping of carbon can increase the surface polarity, electric conductivity, and electron-donor affinity for the carbon network. Moreover, some reports demonstrated that the type of doped N configuration (pyrrolic-N, pyridinic-N, or graphitic-N) has a significant effect on the charge storage process, and the content of the active N in carbon networks could be key parameter determining the activity of the electrode. Among them, pyridinic N has been widely considered as more active for adsorbing alkali metal ions [33]. The capacitance of nitrogen-doped carbons (NPCs) comes not only from EDLC but also from the faradaic reaction at or near the surface of the carbon material [34]. Uppugalla et al. reported a nitrogen and sulfur heteroatom co-doped activated carbon by hydrothermal reaction, which served as an electrode for SCs and delivered an enhanced capacitance of 417 F g^−1^ at 0.7 A g^−1^ with a good cycling stability [35]. Llnicka et al. prepared an N-doped porous carbon from gelatin and green algae, and its nitrogen functionalities and microstructure enabled a capacitance of 327 F g^−1^ with cycle durably [36]. Generally, the preparation of NPCs is achieved through the pyrolysis of biomass waste containing N elements, and the post-treatment of carbon materials with ammonia and urea at high temperatures [37]. Therefore, we anticipated that pyrolyzing the reed straw with N-containing organics would fabricate porous carbon materials with rich active sites, which can serve as the electrode to boost SCs.

In this work, a reed straw derived biomass carbon (RSM) was prepared by directly pyrolyzing reed straw and melamine with KOH as the activator. The RSM as the electrode for SCs exhibited an enhanced electrochemical performance by rationally introducing melamine. In addition, the pyrolysis temperature was taken into consideration. The optimized RSM-0.33-550 (the mass ratio of melamine to reed straw is 1:3 and the optimal activation temperature is 550 °C) with an N content of 6.02% and a specific surface area of 547.1 m^2^ g^−1^ displayed a capacitance of 202.8 and 164 F g^−1^ at a current density of 1 and 20 A g^−1^, respectively. At a high current density of 20 A g^−1^, it still retained a capacitance of 158 F g^−1^. It also showed an outstanding stability with capacity retention of 96.3% at 20 A g^−1^ after 5000 cycles.

## 2. Results and Discussion

Figure 1 shows the schematic illustration of the preparation process of RSM-y-5. The reed straw powder and melamine were firstly mixed as the carbon and the nitrogen resource, respectively. Subsequently, the mixture was further pyrolyzed under an N_2_ atmosphere with KOH as the activator. Finally, the N-doped porous carbon was collected after washing with HCl to remove impurities. Figure 2 shows the SEM images of different RSM-y-550 samples (y = 0, 0.33, 1, and 3). The low-magnification SEM images of all samples show irregular particle morphology, implying a defected structure of carbon material after pyrolysis and the activation process. Figure 2a,b show the SEM images of the RSM-0-550 sample without melamine addition. After introduction of melamine, no obvious change in morphology was observed (Figure 2c–h). Notably, some macropore structures can be found from the high-magnification SEM images for the RSM-0.33-550 (Appendix A) and RSM-1-550 (Appendix A) samples, and such a macropore is conductive to ion transport in the electrolyte. In addition, the effect of activation temperature on the morphology of carbon was also investigated. For simplification, the prepared carbon materials were named as RSM-1-x, and the x represents the value of the pyrolysis temperature of 400, 500, 550, 600, and 650 °C. As shown in Appendix A, the morphology of the carbon material shows more disorder and porous structure with the increase of activation temperature, which implies that a higher pyrolysis temperature is favorable for the formation of porous structures. 

Figure 3a clearly shows the XRD pattern of RSM-y-550. It can be observed that all the XRD patterns show a broad peak at 25°, which corresponds to the typical feature of amorphous carbon [38]. Additionally, no characteristic peak of C_3_N_4_ derived from melamine is found [39,40]. Figure 3b shows the Raman spectra of a series of RSM-y-550 samples. Two obvious peaks are observed at 1363 and 1596 cm^−1^, corresponding to G and D bands, respectively [41]. The D band is related to the graphitic defective and disordered carbon structure, while the G band is associated with the E_2g_ symmetry between graphite layers, which indicates an ordered carbon structure [42]. The intensity ratio of G to D band (I_G_/I_D_) for all samples are illustrated in Figure 3b. With increased melamine addition, I_G_/I_D_ decreased from 1.310 to 1.232, indicating more disorder of the carbon structures. This could be conducive to the diffusion of electrolyte ions and the improvement of EDLC. It is worth noting that the value of I_G_/I_D_ is 1.324 for the RSM-3-550 sample, which suggests that the excessive addition of melamine increased the degree of carbon graphitization. This could be due to the formation of C_3_N_4_ after excessive melamine introduction. In addition, Appendix A shows the XRD patterns of RSM-1-x obtained at different pyrolysis temperatures, and similar results are observed. The Raman spectra of RSM-1-x are displayed in Appendix A. Notably, with the increased activation temperature from 400 to 600 °C, the value of I_G_/I_D_ decreased from 1.466 to 1.185. This indicates the higher temperature is conductive to the activation process, thus providing more interface for electrode and electrolyte [43]. However, the increased value of I_G_/I_D_ of 1.213 is observed for the RSM-1-650 sample, which could be attributed to the enhanced degree of graphitization at a higher pyrolysis temperature. 

The wide scan XPS spectrum of RSM-0-550 (Figure 4a) shows the existence of C, N, and O elements, and the content was measured to be 81.1%, 5.2% and 13.6%, respectively. The existence of the N element is ascribed to the protein from the reed straw [44]. After melamine addition, the obtained RSM-0.33-550 presents an N content of 6.1%, suggesting N introduction (Figure 4d) [45]. Figure 4b,e compare the C 1s core-level XPS spectra of the RSM-0-550 and RSM-0.33-550 samples. Four peaks at 284.8, 286.1, 287.6, and 289.2 eV are observed, corresponding to the C-C/C=C, C-N/C-O, C=O, and O=C-O, respectively [46,47]. Notably, a small decline in the content of C-C and a small increase in C-N are observed for RSM-0.33-550 compared with RSM-0-550, implying that a part of the C-N bond replaced the C-C bond after melamine introduction. The N 1s XPS spectra of RSM-0-550 (Figure 4c) and RSM-0.33-550 (Figure 4f) display the four peaks at 398.6, 400.1, 401.3 and 405.6 eV, corresponding to the Pyridinic-N (N-6), Pyrrolic-N (N-5), Graphitic-N (G-N), and N-Oxide (N-O), respectively [15,48]. The relative densities of these N species can be illustrated from the proportional areas of corresponding peaks as displayed in Appendix A. It is well known that the N-6 is more active compared with the other types of N, due to it having more active sties and sufficient defects, and it is more energetically favorable to promoting the interaction between the electrode and electrolyte ions (alkali metal ions) [33]. Clearly, the content of N-6 increased from 15.4% to 24.8% after melamine introdution, indicating the increased active sites for charge storage. Moreover, the decline content of G-N from 14.9% to 10.3% after melamine introduction implies more exposed edge sites, which can be due to the pyrolysis and activation process of melamine [49]. The N doping can considerably boost the performance of SCs by tuning the electronic conductivity, surface accessibility, and faradaic reaction. In the carbon framework, the N and C atoms with multi-graded electronegativities give a polarized/accessible electrode surface with enhanced electroabsorption active sites for improved electrolyte–electrode interaction [50]. Additionally, the boosted adsorption of alkali metal ions can be achieved at the N faradaic-active sites, resulting in good pseudocapacitance by electrochemical redox reactions [50,51].

To study the porosity of the carbon materials before and after melamine addition, BET measurement was conducted. As displayed in Figure 5a,c, N_2_ adsorption–desorption isotherms of RSM-0-550 and RSM-0.33-550 display a sharply rising adsorption quantity at P/P_0_ < 0.05, indicating a type I (IUPAC classification) feature for microporous materials [52]. Clearly, the obvious hysteresis loop at P/P_0_ = 0.45–0.9 and sharply increased at P/P_0_ = 0.9–1.0, indicates the existence of both meso- and microporous stricture in the prepared RSM-0-550 and RSM-0.33-550 samples [49]. Figure 5b,d shows the corresponding pore size distribution, and mesopores and micropores with diameters of around 1.0 and 2.3 nm are achieved, respectively. The mesoporous pores have a positive effect on the charge storage, which is important for reaching high capacitance [42]. The detailed parameters are shown in Appendix A, and it shows that the surface area of RSM-0-550 is 514.2 m^2^ g^−1^ with a pore volume of 0.141 cm^3^ g^−1^. After melamine addition, the surface area of 547.1 m^2^ g^−1^ with a pore volume of 0.159 cm^3^ g^−1^ was achieved for RSM-0.33-550. It is concluded that the addition of nitrogen source is not the important factor in generating the porous structure, and the surface area is mainly ascribed to the amount of activator and the pyrolysis temperature [42]. Clearly, RSM-0.33-550 with hierarchical porosity can give an abundant surface area and rapid electrolyte ion transfer for energy storage.

The electrochemical performance of all samples was evaluated by a three-electrode system, in which 6 M KOH was employed as the electrolyte due to its a higher ionic conductivity and the smaller radius of the hydration sphere (3.00 Å for OH^−^ and 3.31 Å for K^+^). Appendix A shows the GCD curves of RSM-1-x prepared by the different pyrolysis temperatures. Clearly, the RSM-1-400, RSM-1-500, RSM-1-550, RSM-1-600, and RSM-1-650 electrodes delivered a capacitance of 45.6, 145.4, 164.7, 146.3, and 120.1 F g^−1^, respectively. Based on the GCD data, the related rate performance was illustrated in Appendix A. In addition, Appendix A illustrates the CV curves of the RSM-1-y electrode and similar results were found. As shown in Appendix A, the capacitance of 42.3, 131.2, 153.4, 142, and 114.9 F g^−1^ was achieved for RSM-1-400, RSM-1-500, RSM-1-550, RSM-1-600, and RSM-1-650 electrodes, respectively. Notably, the CV curves of RSM-1-550, RSM-1-600, and RSM-1-650 electrodes (Appendix A) show a relative quasi-rectangle shape compared with the RSM-1-400 and RSM-1-500, which is ascribed to a higher pyrolysis temperature providing a sufficient activation process, thus offering capacitive behavior for charge storage. This indicates that the activation temperature has a great influence on the electrochemical performance, which is mainly ascribed to the change in the structure and surface chemistry of carbon materials.

Figure 6a summarizes the GCD curve of the RSM-y-550 electrode at a current density of 1 A g^−1^. All GCD curves present a relatively symmetrical profile, indicating good coulombic efficiency and capacitive properties. Notably, the RSM-0.33-550 electrode exhibited a longer discharge time, which implies it has a superior capacitance. The specific capacitance of RSM-0-550, RSM-0.33-550, RSM-1-550, and RSM-3-550 was calculated to be 181.6, 202.8, 164.7, and 159.3 F g^−1^, respectively. This indicates that the rational melamine introduction is beneficial in improving the electrochemical performance. The GCD curves of RSM-y-550 at different current densities are shown in Figure 6b and Appendix A. All GCD curves show a symmetrical triangular shape, indicating good capacitive behavior. The corresponding rate performance based on GCD data is displayed in Figure 6c. Clearly, RSM-0.33-550 delivered a superior rate performance with a capacitance of 202.8, 194.4, 185.5, and 177 F g^−1^ at 1, 2, 5, and 10 A g^−1^. Even at a current density of 20 A g^−1^, a capacitance of 164 mF g^−1^ can be achieved, which is higher than that of RSM-0-550 (148 F g^−1^), RSM-0-550 (120 F g^−1^), and RSM-0-550 (98 F g^−1^). In addition, the GCD curves of RSM-y-550 were also given in Figure 6d and Appendix A. The capacitance of 171.2, 189.7, 153.4, and 146.3 F g^−1^ was reached for RSM-0-550, RSM-0.33-550, RSM-1-550, and RSM-3-550, respectively. Appendix A compares the electrochemical performance of RSM-0.33-550 and the reported carbonaceous materials electrode in supercapacitors. The RMS-0.33-550 with a specific capacitance of 202.8 F g^−1^ at 1 A g^−1^ exhibits a satisfactory performance for SCs, which is mainly ascribed to a micro- and meso-porous structure and the rich active nitrogen functional group.

Furthermore, EIS measurement was employed as shown in Figure 6e. The Nyquist plots of RSM-0-550, RSM-0.33-550, and RSM-1-550 displayed a small semi-circle in the high frequency region, indicating their low charge transport resistance [53]. On the contrary, the Nyquist plot of RSM-3-550 presents a larger semicircle in the high frequency region. The increased charge transport resistance could be ascribed to the formation of C_3_N_4_ derived from melamine after introducing the excessive N resource [54]. In the low frequency region, it can be seen that RSM-0.33-550 has the highest slope, indicating that the electrolyte ion diffusion of RSM-0.33-550 is the most active and can provide faster current response. Figure 6f shows the long cycling test of the RSM-0.33-550 electrode. Notably, it exhibited good cycle stability with a capacitance retention rate of 96.3% (158 F g^−1^) at a current density of 20 A g^−1^ after 5000 cycles.

## 3. Experimental Section

### 3.1. Materials and Methods

Reed straw powder (60 mesh) was purchased from Shaanxi Jinhe Agricultural Technology Co., Ltd. (Shaanxi, China). Melamine and potassium hydroxide (KOH) were purchased from Shanghai Aladdin Biochemical Technology Co., (Shanghai, China). Polyvinylidene fluoride (PVDF), acetylene black, concentrated hydrochloric acid (HCl), N-methylpyrrolidone (NMP), and other chemical reagents were purchased from Chengdu Kelong Chemical Reagent Co., (Chengdu, China). Nitrogen was purchased from Guangdu Gas Business Department (Chengdu, China). All reagents were analytical grade and used as received without further purification.

### 3.2. Preparation of RSM-y-550

An aliquot of 2 g of reed straw powder was mixed with varying weights of melamine (0 g, 0.66 g, 2 g, and 6 g), and stirred continuously with a glass rod after the addition of 50 mL deionized water until the straw powder no longer floated at the liquid level and was well dispersed in the water. It was then transferred to a 75 mL Teflon stainless steel autoclave and reacted in an oven at 120 °C for 2 h to allow good mixing and make it partially carbonize. When cooled to room temperature, a yellowish powder (RSM) was obtained. Subsequently, 1 g of RSM powder and 1 g of KOH were added to 10 mL deionized water with continuous stirring for 4 h, and then dried at 80 °C for 12 h. The mixture was transferred in a porcelain boat and put in a tubular furnace for further activation. The activator KOH can react with the carbon precursor at the elevated temperature to form a porous structure. The generated CO, CO_2_, and K could further etch the carbon network to form the micro- and meso-porous structures. The activation operation was conducted at 550 °C for 2 h, with a temperature rising rate of 5 °C min^−1^ under a nitrogen atmosphere. After cooling down to room temperature, the mixture was collected and cleaned with 1 M hydrochloric acid and washed with distilled water several times to remove impurities including K_2_CO_3_, KHCO_3_, and KOH. Finally, the products were dried by vacuum at 60 °C for 12 h. The obtained sample was denoted RSM-y-550, where y represents the mass ratio of melamine to reed straw (y = 0, 0.33, 1, and 3).

### 3.3. Characterization

XRD patterns were collected on an X-ray diffractometer (Philip Company, Pw1730) equipped with a Cu Kα radiation (λ = 1.5418 Å). XPS spectra were measured on an X-ray photoelectron spectrometer (XPS, a Kratos Axis Ultra spectrometer, Kratos Analytical Inc., Manchester, UK). SEM images were captured on a Hitachi S-4800 field emission scanning electron microscope (Hitachi High-Tech Co., Tokyo, Japan) at an accelerating voltage of 20 kV. Raman spectra were measured on a DXR Raman microscope (Thermo Fisher Scientific Inc., Waltham, MA, USA) with an excitation wavelength of 455 nm. Brunauer–Emmett–Teller (BET) surface area was collected on a Micromeritics TRISTAR II3020 surface area analyzer (Micromeritics Instrument, Norcross, GA, USA).

### 3.4. Electrochemical Measurement

To prepare the RSM-y-550 working electrode, the prepared mixture of active carbon material (80 wt%), polyvinylidene fluoride (10 wt%), and acetylene black (10 wt%) was coated on nickel foam (1.0 × 1.0 cm^2^) and dried at 80 °C for 24 h. The three-electrode system was employed to estimate the electrochemical performance of all electrodes. RSM-y-550 was used as the working electrode (anode), with platinum as the opposite electrode, saturated calomel electrode (SCE) as the reference electrode, and 6 M KOH solution as the electrolyte. Cyclic voltammetry (CV), galvanostatic charge–discharge (GCD) and electrochemical impedance spectroscopy (EIS) were performed on a CHI600E electrochemical workstation (Chenhua Instrument Co., Shanghai, China). The GCD was employed at a current density of 1–10 A g^−1^. The CV was conducted at a scan rate of 10–100 mV s^−1^. The frequency range of EIS was 0.01–100 kHz, and the potential window was 5 mV. According to the results of the GCD curve, the capacitance was calculated as follows: C = I × ∆t/(m × ∆V), where ∆V (V) is the potential window, m (g) is the mass load of the active substance, ∆t (s) is the discharge time, I (A) is the current density, C (F g^−1^) is the specific capacitance. 

## 4. Conclusions

In summary, we proposed an efficient route to prepare an N-doped carbon material by directly pyrolyzing reed straw and melamine. The pyrolysis temperature and the amount of activator were carefully investigated. The rational regulation of the activation temperature and the amount of the N resource was significant for the electrochemical performance of the reed-straw based electrode. The optimized RSM-0.33-550 prepared at 550 °C with a mass ratio of melamine to reed straw of 0.33 possessed an N content of 6.02% and a specific surface area of 547.1 m^2^ g^−1^. As the anode for SCs, the RSM-0.33-550 electrode delivered a higher capacitance of 202.8 F g^−1^ at a current density of 1 A g^−1^. EIS measurement exhibited a faster ion transfer for the RSM-0.33-550 electrode. Even at a current density of 20 A g^−1^, the RSM-0.33-550 electrode can achieve a capacitance of 164 mF g^−1^, which is higher than that of RSM-0-550 (148 F g^−1^), RSM-0-550 (120 F g^−1^), and RSM-0-550 (98 F g^−1^). Notably, the RSM-0.33-550 electrode showed an excellent cyclic stability with 96.3% capacitance retention after 5000 cycles. This work offers a new insight into rationally utilizing biomass waste to prepare functionalized carbon materials and their related application.

## Figures and Tables

**Figure 1 molecules-28-04633-f001:**
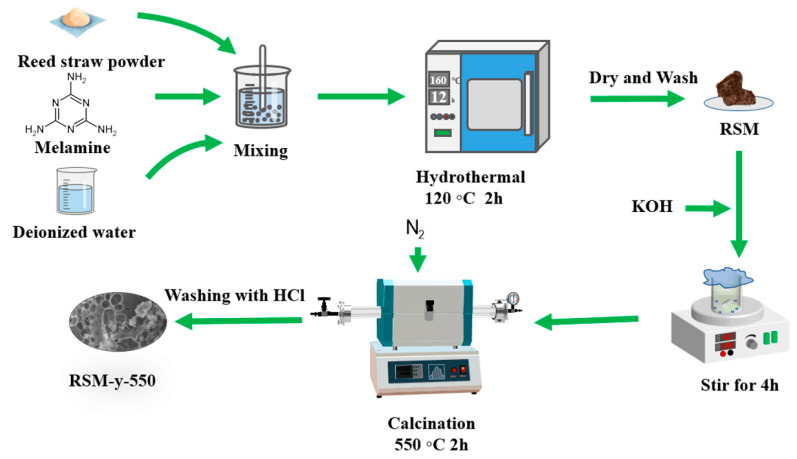
Schematic illustration of the synthesis procedure of RSM-y-5.

**Figure 2 molecules-28-04633-f002:**
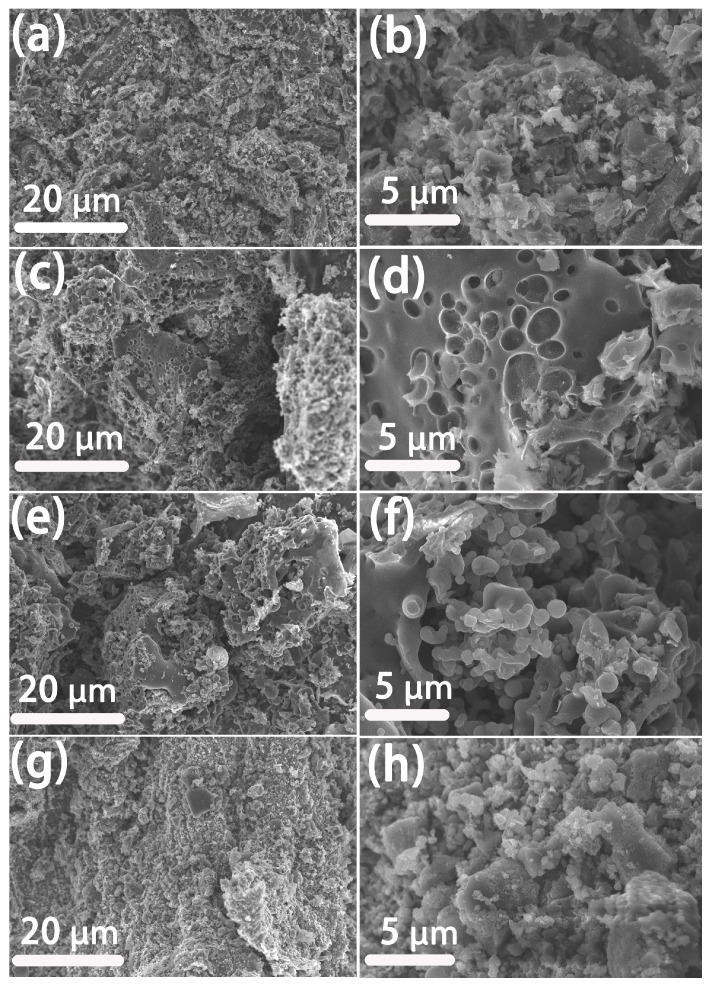
SEM image of reed straw based carbon material: (**a**,**b**) RSM-0-550, (**c**,**d**) RSM-0.33-550, (**e**,**f**) RSM-1-550, and (**g**,**h**) RSM-3-550.

**Figure 3 molecules-28-04633-f003:**
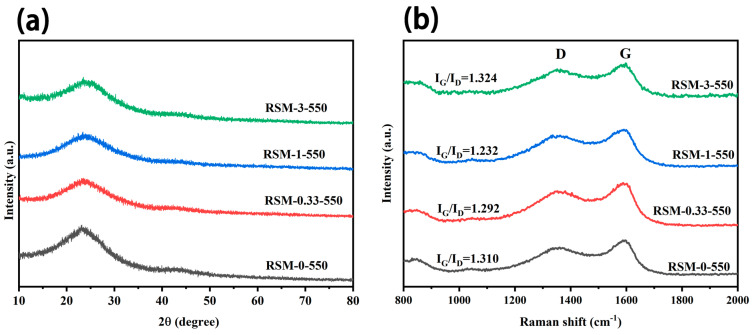
(**a**) XRD patterns of RSM-y-550 (y = 0, 0.33, 1, and 3), and (**b**) Raman spectra of RSM-y-550 (y = 0, 0.33, 1, and 3).

**Figure 4 molecules-28-04633-f004:**
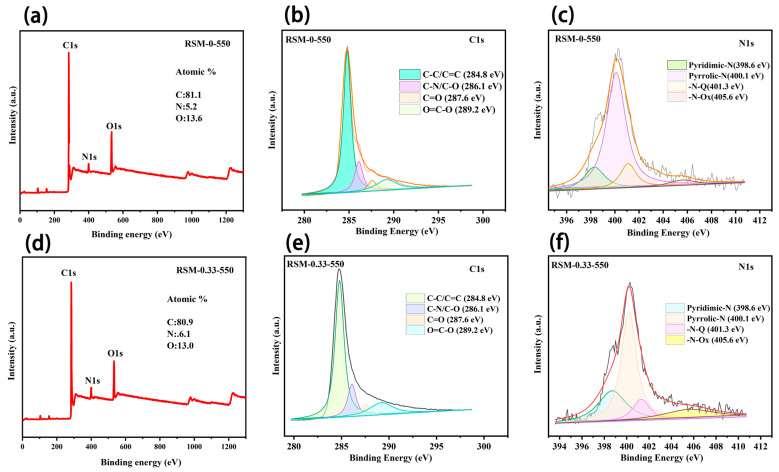
Wide scan XPS spectrum of (**a**) RSM-0-550. XPS spectra of RSM-0-550 for (**b**) C 1s and (**c**) N 1s regions. Wide scan XPS spectrum of (**d**) RSM-0.33-550. XPS spectrum of RSM-0.33-550 for (**e**) C 1s and (**f**) N 1s regions.

**Figure 5 molecules-28-04633-f005:**
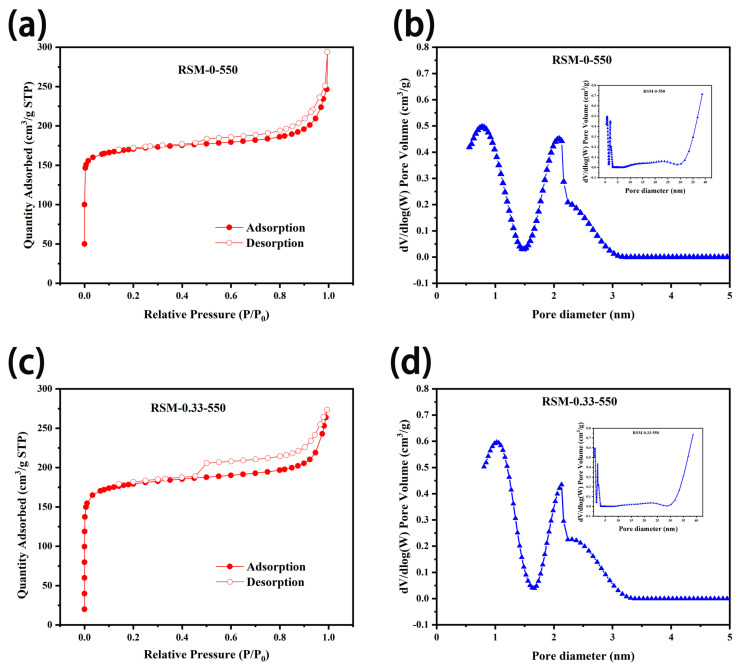
N_2_ adsorption and desorption isothermal curves of (**a**) RSM-0-550 and (**c**) RSM-0.33-550, and the corresponding pore size distributions for (**b**) RSM-0-550 and (**d**) RSM-0.33-550.

**Figure 6 molecules-28-04633-f006:**
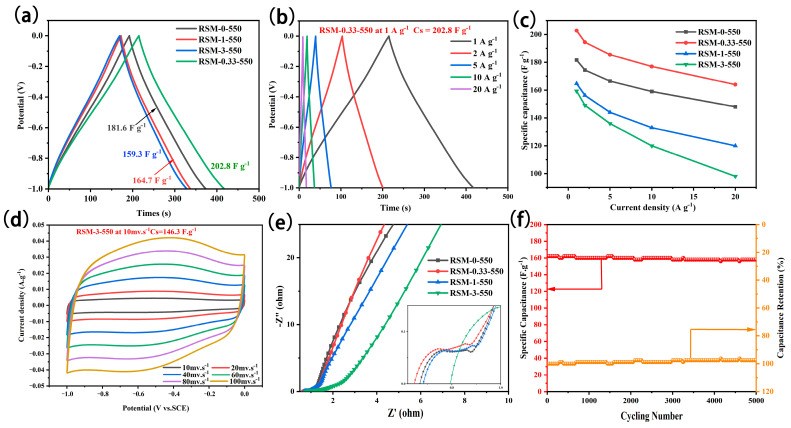
(**a**) GCD curves of all RSM-y-550 (y = 0, 0.33, 1, and 3) electrodes at a current density of 1 A g^−1^. (**b**) GCD curves of RSM-0.33-550 at the current densities of 1, 2, 5, 10, and 20 A g^−1^. (**c**) Rate performance at the current density of 1, 2, 4, 5, 8 and 10 A g^−1^. (**d**) CV curves of RSM-0.33-550 at various scan rates of 10–150 mV s^−1^. (**e**) Nyquist plots of RSM-y-550 (y = 0, 0.33, 1, and 3). (**f**) Long cycling test at a current density of 20 A g^−1^ after 5000 cycles.

## Data Availability

Data are contained within the article.

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
