# Peer review of "Biomass Derived N-Doped Porous Carbon Made from Reed Straw for an Enhanced Supercapacitor"

_molecules, 2023, doi:10.3390/molecules28124633_

Round 1

Reviewer 1 Report

This manuscript describes the Reed straw-derived N-doped carbon for electric double-layer capacitor applications. This manuscript can be of interest to readers in this community. The details are written below. It needs some rectifications before one can take a final decision.

1.     Introduction could be strengthened: (i) The is no clear motivation for the study, (ii) Novelty is not clear, (iii) The literature is not very adequately discussed, for example, there are many recent research papers (ex: DOI: 10.1080/21663831.2022.2139163 and 10.1007/s42247-023-00503-1) on N and/S co-doping, discuss the systems used and the findings more data should be added in tabular form to compare data with literature findings in similar mediums.

2.     Compare present work with these reported papers (DOI: 10.1016/j.matlet.2017.02.007 and 10.1007/s10008-015-3061-y)

3.     Provide the surface area of the samples RSM-y-550 (y = 0, 1, and 3). Explain and discuss the difference in surface area of doped materials.

4.     Page 7, line 227: include current density value.

5.     No scientific explanations are given to observations, and no reference are made to literature in the results and discussions.

6.     The authors should include a Ragone plot to compare the performance of these works to others in the literature.

Minor editing of English language required

Author Response

Reviewer 1:

This manuscript describes the Reed straw-derived N-doped carbon for electric double-layer capacitor applications. This manuscript can be of interest to readers in this community. The details are written below. It needs some rectifications before one can take a final decision.

Response:

We appreciate for the reviewer’s good suggestion on revising our manuscript. The manuscript has been carefully revised following the reviewer’s suggestion accordingly.

1. Introduction could be strengthened: (i) The is no clear motivation for the study, (ii) Novelty is not clear, (iii) The literature is not very adequately discussed, for example, there are many recent research papers (ex: DOI: 10.1080/21663831.2022.2139163 and 10.1007/s42247-023-00503-1) on N and/S co-doping, discuss the systems used and the findings more data should be added in tabular form to compare data with literature findings in similar mediums.

Response:

We appreciated for the review’s good comments. As suggested, we studied the above literature, and the related discussion was added in introduction section. (Please see Ref. 35 and 36, line 95-99).

2. Compare present work with these reported papers (DOI: 10.1016/j.matlet.2017.02.007 and 10.1007/s10008-015-3061-y)

Response:

As suggested, the related discussion about the above reported papers was added in introduction section. (Please see page 2, line 70-76)

3. Provide the surface area of the samples RSM-y-550 (y = 0, 1, and 3). Explain and discuss the difference in surface area of doped materials.

Response:

We appreciated for the review’s good comments. As suggested, we updated Figure 5 and added more related description. Due to the limitation of equipment and time, we only compared the N2 adsorption and desorption isothermal curves and the corresponding pore size distributions for RSM-0-550 and RSM-0.33-550 samples. As shown in Figure 5, similar N2 adsorption and desorption isothermal curves can be observed. The surface area of RSM-0-550 and RSM-0.33-550 was measured to be 514.2 and 547.1 m2 g-1, respectively. (Please see page 7, line 250-264)

4. Page 7, line 227: include current density value.

Response:

As suggested, we added the related description in the caption of Figure 6. (Please see page 8, line 268-269)

5.No scientific explanations are given to observations, and no reference are made to literature in the results and discussions.

Response:

We appreciated for the review’s good comments. As suggested, we added some citations in results and discussion section. (Please see Ref. 42, 43 and 50; page 7, line 262-266).

6. The authors should include a Ragone plot to compare the performance of these works to others in the literature.

Response:

We appreciated for the review’s good comments. Ragone plot is usually based on the GCD data of two-electrode system including asymmetric and symmetric supercapacitors. Commonly, carbon materials as the anode for SCs was evaluated in 6 M KOH with the potential in a range of -1.0–0 V. Therefore, to compare RSM-0.33-550 with previous electrodes, we listed a table as shown in Table S2 in supporting information. We hope this comparison can present the good electrochemical performance of such RSM-0.33-550 to a certain extent.

Reviewer 2 Report

In this paper, the authors reported a biomass-derived N-doped porous carbon made from reed straw, which was used as electrodes for capacitors. The carbon electrodes showed good capacitive performance with a capacitance of 202.8 F/g at 1 A/g and good cycling performance. This paper is well-written and could be accepted after a minor revision.

1.  What is the difference of the N doping between RSM-0-550 and RSM-0.33-550? In figure 4, they have similar content of N heteroatoms with similar component.

2. Please provide the pore diameter over 5 nm in Figure 5b.

3. The scale bars in Figure 2 are not clear.

4. Some important references should be cited to improve this paper: e.g., Tungsten. 2023, 5(1):118-129; J. Mater. Chem. A, 2022,10, 9612-9620; chemical engineering journal, 2022, 449, 137561.

 Minor editing of English language required

Author Response

Reviewer 2:

In this paper, the authors reported a biomass-derived N-doped porous carbon made from reed straw, which was used as electrodes for capacitors. The carbon electrodes showed good capacitive performance with a capacitance of 202.8 F/g at 1 A/g and good cycling performance. This paper is well-written and could be accepted after a minor revision.

Response:

We appreciate for the reviewer’s good suggestion on revising our manuscript. The manuscript has been carefully revised following the reviewer’s suggestion accordingly.

1. What is the difference of the N doping between RSM-0-550 and RSM-0.33-550? In figure 4, they have similar content of N heteroatoms with similar component.

Response:

We appreciate for the reviewer’s good suggestion on revising our manuscript. Although the RSM-0-550 and RSM-0.33-550 samples showed a similar N content, but RSM-0.33-550 displayed a more content of “active N”. It is well known that the N-6 presents more active compared with the other types of N, due to it can give more active sties and sufficient defects, and is more energetically favorable towards promoting the interaction between the electrode and electrolyte ions (alkali metal ions). As shown in Figure S4, the content of N-6 increased from 15.4% to 24.8% after melamine introducing, indicating the increased active sites for charge storage.

2. Please provide the pore diameter over 5 nm in Figure 5b.

Response:

We appreciated for the review’s good comments. As suggested, we carefully rectified the Figure 5. (Please see page 8, Figure 5)

3. The scale bars in Figure 2 are not clear.

Response:

We appreciated for the review’s good comments. As suggested, we carefully rectified the scale bars. (Please see page 5, Figure 2)

4. Some important references should be cited to improve this paper: e.g., Tungsten. 2023, 5(1):118-129; J. Mater. Chem. A, 2022,10, 9612-9620; chemical engineering journal, 2022, 449, 137561.

Response:

As suggested, the references were cited accordingly. (Please see Ref. 5, 43, and 12).

Reviewer 3 Report

This work presents an investigation into the development of N-doped porous carbon derived from reed straw and melamine as a means to enhance supercapacitor performance. The authors have chosen an approach by utilizing biomass waste as a precursor for advanced carbon materials, which aligns with the growing interest in sustainable and environmentally friendly solutions.

The following comments need to be considered by the authors before recommending the article for publication:

While the abstract mentions that developing advanced carbon materials from biomass waste has attracted attention, it would be helpful to highlight the specific novelty or contribution of this study. What sets the N-doped carbon material (RSM-0.33-550) apart from previous works? Emphasizing the unique aspects of the material or the approach used in its synthesis will make the abstract more appealing to readers.

Emphasize the significance and implications of the obtained results. Discuss how the reported capacitance, stability, and capacity retention values compare to existing carbon materials for supercapacitor applications. Highlight the potential of the N-doped carbon material (RSM-0.33-550) as an electrode material for supercapacitors and its suitability for energy storage applications.

The introduction lacks a clear focus on the specific problem or research gap that the study aims to address. It should provide a stronger rationale for why developing energy storage devices using biomass waste is important in the context of the current challenges caused by fossil fuel combustion.

While some general information is provided about electric double-layer capacitors (EDLCs) and biomass carbon materials, the introduction lacks sufficient context and background information on the current state of research in the field. It should include relevant studies and advancements in the area of biomass-derived carbon materials for supercapacitors.

Begin the introduction by clearly identifying the research gap and the specific objective of the study. This will help readers understand the significance of the research and its contribution to the existing knowledge.

Include a concise yet comprehensive literature review that highlights key studies and advancements in the field of biomass-derived carbon materials for supercapacitors. Discuss the limitations of existing materials and emphasize the need for improved capacitance and energy density.

Although reed straw is mentioned as a potential precursor for carbon materials, the introduction does not adequately justify why reed straw was chosen or highlight its unique properties compared to other biomass sources. It should provide a stronger rationale for selecting reed straw as the precursor material and explain how its specific characteristics contribute to enhancing the faradaic reaction and specific capacitance.

Highlight the role of nitrogen doping: Emphasize the role of nitrogen doping in enhancing the capacitance and energy storage capabilities of carbon materials. Discuss the benefits of nitrogen-containing functional groups and their impact on surface polarity, electric conductivity, and electron-donor affinity.

Clearly describe the experimental approach used to prepare the biomass-derived carbon material, including the specific pyrolysis process, the role of melamine, and the optimization of parameters such as the mass ratio and activation temperature. This will provide readers with a better understanding of the methodology and the rationale behind the chosen conditions.

Organize the introduction in a logical and coherent manner, ensuring smooth transitions between different sections. Consider dividing it into subsections to address specific aspects, such as the importance of energy storage devices, the potential of biomass-derived carbon materials, the role of nitrogen doping, and the specific objectives of the study.

In the experimental section, explain the rationale behind selecting specific reaction conditions such as temperature, time, and reagent amounts. Refer to previous studies or preliminary experiments that support these choices.

Include details on the stirring speed, duration, and intensity during the mixing process of reed straw powder, melamine, and deionized water. This will help ensure uniform mixing and improve reproducibility.

Clarify the purpose and significance of the autoclave reaction at 120 °C for 2 hours. Explain how this step contributes to the formation or modification of the desired product.

Explain the importance of conducting activation at 550 °C for 2 hours. Justify the choice of these specific conditions and describe their impact on the final product.

Provide more details on the purpose and procedure of the sample purification step, including the rationale for using 1 M hydrochloric acid and the number of washing cycles performed.

In Figure 2 include high-resolution images to capture more detailed morphology information beyond the micron scale level.

Author Response

Reviewer 3:

This work presents an investigation into the development of N-doped porous carbon derived from reed straw and melamine as a means to enhance supercapacitor performance. The authors have chosen an approach by utilizing biomass waste as a precursor for advanced carbon materials, which aligns with the growing interest in sustainable and environmentally friendly solutions. The following comments need to be considered by the authors before recommending the article for publication:

Response:

We appreciate for the reviewer’s good suggestion on revising our manuscript. The manuscript has been carefully revised following the reviewer’s suggestion accordingly.

1. While the abstract mentions that developing advanced carbon materials from biomass waste has attracted attention, it would be helpful to highlight the specific novelty or contribution of this study. What sets the N-doped carbon material (RSM-0.33-550) apart from previous works? Emphasizing the unique aspects of the material or the approach used in its synthesis will make the abstract more appealing to readers.

Response:

As suggested, we updated the abstract. (Please see page 1, line 11-20)

2. Emphasize the significance and implications of the obtained results. Discuss how the reported capacitance, stability, and capacity retention values compare to existing carbon materials for supercapacitor applications. Highlight the potential of the N-doped carbon material (RSM-0.33-550) as an electrode material for supercapacitors and its suitability for energy storage applications.

Response:

We appreciated for the review’s comments. As suggested, we added the comparison to reported carbon materials for supercapacitor applications in the supplementary material. The related discussion was added. (Please see page 9, line 304-308)

3. The introduction lacks a clear focus on the specific problem or research gap that the study aims to address. It should provide a stronger rationale for why developing energy storage devices using biomass waste is important in the context of the current challenges caused by fossil fuel combustion.

Response:

We appreciated for the review’s comments. As suggested, the related description was added. (Please see page 2, line 48-57

4. While some general information is provided about electric double-layer capacitors (EDLCs) and biomass carbon materials, the introduction lacks sufficient context and background information on the current state of research in the field. It should include relevant studies and advancements in the area of biomass-derived carbon materials for supercapacitors.

Response:

As suggested, we have intensified the discussion of the biomass carbon and their background of application. (Please see page 1, line 31-40, and 48-51).

5. Begin the introduction by clearly identifying the research gap and the specific objective of the study. This will help readers understand the significance of the research and its contribution to the existing knowledge.

Response:

We appreciated for the review’s comments. As suggested, some recent reports about heteroatoms doping carbon materials and biomass carbon materials as the electrode for supercapacitors were added in the introduction section. (Please see page 2, line 71-77, 95-99)

6. Include a concise yet comprehensive literature review that highlights key studies and advancements in the field of biomass-derived carbon materials for supercapacitors. Discuss the limitations of existing materials and emphasize the need for improved capacitance and energy density.

Response:

We appreciated for the review’s comments. As suggested, the related description was updated. (Please see page 1, line 43-45)

7. Although reed straw is mentioned as a potential precursor for carbon materials, the introduction does not adequately justify why reed straw was chosen or highlight its unique properties compared to other biomass sources. It should provide a stronger rationale for selecting reed straw as the precursor material and explain how its specific characteristics contribute to enhancing the faradaic reaction and specific capacitance.

Response:

We appreciated for the review’s comments. Compared with other biomass waste (peel, shell, bacteria, etc.), reed straw possesses rich cellulose, lignin, and hemicellulose. In addition, about 900 million tons of corn, wheat, rice and sorghum straw are produced every year in China, which can provide sustainable and abundant raw materials for preparing active carbon. Therefore, it is attractive to convert this biomass waste into high-performance and low-cost carbon materials for energy storage. As suggested, the related description was added. (Please see page 2, line 65-69)

8. Highlight the role of nitrogen doping: Emphasize the role of nitrogen doping in enhancing the capacitance and energy storage capabilities of carbon materials. Discuss the benefits of nitrogen-containing functional groups and their impact on surface polarity, electric conductivity, and electron-donor affinity.

Response:

Nitrogen doping of carbon can increase the surface polarity, electric conductivity, and electron-donor affinity for carbon network. The type of doped N configuration (pyrrolic-N, pyridinic-N, or graphitic-N) has a significant role in the charge storage process, and the content of the active N in carbon network could be the key parameter determining the activity of electrode. Among them, pyridinic N is considered as more active for adsorbing alkali metal ions. As suggested, we added the related description accordingly. (Please see page 2, line 85-93)

9. Clearly describe the experimental approach used to prepare the biomass-derived carbon material, including the specific pyrolysis process, the role of melamine, and the optimization of parameters such as the mass ratio and activation temperature. This will provide readers with a better understanding of the methodology and the rationale behind the chosen conditions.

Response:

As suggested, we updated the preparation of RSM-y-550 section.

10. Organize the introduction in a logical and coherent manner, ensuring smooth transitions between different sections. Consider dividing it into subsections to address specific aspects, such as the importance of energy storage devices, the potential of biomass-derived carbon materials, the role of nitrogen doping, and the specific objectives of the study.

Response:

We appreciated for the review’s comments. According to the requirements of periodical format, we think the current format of introduction can be well expressed.

11. In the experimental section, explain the rationale behind selecting specific reaction conditions such as temperature, time, and reagent amounts. Refer to previous studies or preliminary experiments that support these choices.

Response:

As suggested, the related description was added. (Please see page 3, line 126-141).

12. Include details on the stirring speed, duration, and intensity during the mixing process of reed straw powder, melamine, and deionized water. This will help ensure uniform mixing and improve reproducibility.

Response:

As suggested, we added the detailed description in experiment section (Please see page 3, line 126-128).

13. Clarify the purpose and significance of the autoclave reaction at 120 °C for 2 hours. Explain how this step contributes to the formation or modification of the desired product.

Response:

The mixture was transferred to a 75 mL Teflon stainless steel autoclave and reacted in an oven at 120 °C for 2 h to mix the mixture well and make it partially carbonize. This process allows the reed straw to be fully mixed with melamine. As suggested, we added the detailed description in experiment section. (Please see page 3, line 129)

14. Explain the importance of conducting activation at 550 °C for 2 hours. Justify the choice of these specific conditions and describe their impact on the final product.

Response:

In our work, the temperature was initially studied. The activation temperature of 400, 500, 550, 600, and 650 °C. The corresponding SEM, XRD, Raman, and electrochemical performance were illustrated in Supplementary material. Notably, as shown in Figure S5f, when the pyrolysis temperature reached 550 °C, the obtained carbon electrode delivered a higher specific capacitance. This can be ascribed to the formation of porous structure by using KOH as the activator, and the results is consistence with the previous reports (Journal of Materials Science & Technology 2023, 134, 142-150. Journal of Power Sources 2022, 517, 230727.). In addition, we also gave the BET data of RSM-0-550. The result indicates that the N resource could be not the factor for the formation of porous structure. Therefore, the activation process, especially the activation temperature is significant for the fabrication of such RSM-0.33-550.

15. Provide more details on the purpose and procedure of the sample purification step, including the rationale for using 1 M hydrochloric acid and the number of washing cycles performed.

Response:

The hydrochloric acid was employe to wash the carbon materials for removing the possible by-product K2CO3, KHCO3, and KOH. As suggested, the detailed description was added. (Please see page 3, line 138-140)

16. In Figure 2 include high-resolution images to capture more detailed morphology information beyond the micron scale level.

Response:

We appreciated for the review’s good comments. As suggested, we added the high-magnification SEM images of RSM-1-550 and RSM-0.33-550 in the supplementary material. (Please see Figure S1). In addition, we also updated the Figure 2 and S2 to make the image clearer.

Round 2

Reviewer 1 Report

Accept in it's present form.

Minor editing required

Reviewer 3 Report

The manuscript can be considered for acceptance in its present form.